# OGT Binding Peptide-Tagged Strategy Increases Protein O-GlcNAcylation Level in *E. coli*

**DOI:** 10.3390/molecules28052129

**Published:** 2023-02-24

**Authors:** Yang Li, Zelan Yang, Jia Chen, Yihao Chen, Chengji Jiang, Tao Zhong, Yanting Su, Yi Liang, Hui Sun

**Affiliations:** 1College of Life Sciences, Wuhan University, Wuhan 430072, China; 2School of Basic Medical Sciences, Xianning Medical College, Hubei University of Science and Technology, Xianning 437100, China; 3Taikang Center for Life and Medical Sciences, Hubei Key Laboratory of Cell Homeostasis, Wuhan University, Wuhan 430072, China; 4Hubei Province Key Laboratory of Allergy and Immunology, Wuhan University, Wuhan 430072, China

**Keywords:** OGT, O-GlcNAc, Tau, OBP-tagged strategy

## Abstract

O-GlcNAcylation is a single glycosylation of GlcNAc mediated by OGT, which regulates the function of substrate proteins and is closely related to many diseases. However, a large number of O-GlcNAc-modified target proteins are costly, inefficient, and complicated to prepare. In this study, an OGT binding peptide (OBP)-tagged strategy for improving the proportion of O-GlcNAc modification was established successfully in *E. coli*. OBP (P1, P2, or P3) was fused with target protein Tau as tagged Tau. Tau or tagged Tau was co-constructed with OGT into a vector expressed in *E. coli*. Compared with Tau, the O-GlcNAc level of P1Tau and TauP1 increased 4~6-fold. Moreover, the P1Tau and TauP1 increased the O-GlcNAc-modified homogeneity. The high O-GlcNAcylation on P1Tau resulted in a significantly slower aggregation rate than Tau in vitro. This strategy was also used successfully to increase the O-GlcNAc level of c-Myc and H2B. These results indicated that the OBP-tagged strategy was a successful approach to improve the O-GlcNAcylation of a target protein for further functional research.

## 1. Introduction

O-GlcNAcylation is a highly dynamic and reversible PTM in the serine/threonine residues of nuclear, cytosolic, and mitochondrial proteins [1,2]. Only one O-GlcNAc transferase (OGT) catalyzes the addition of O-GlcNAc and one O-GlcNAcase (OGA) removes the modification [3,4,5]. Mounting evidence has shown that O-GlcNAcylation regulates essential cellular processes such as transcription, metabolic regulation, signal transduction, cell cycle, nutrient sensing, protein stabilization, and cellular responses to diverse stress conditions [6,7,8]. Aberrant O-GlcNAcylation has been implicated in the progression of a wide range of diseases, such as diabetes [9], cancer [10], and cardiovascular [11] and neurodegenerative diseases [12]. In humans, OGT is encoded by the OGT gene on the X chromosome (Xq13.1) [13], and consists of an N-terminal tetratricopeptide repeat (TPR) and a C-terminal catalytic domain [14]. TPR repeats are mainly involved in the binding of OGT to substrate proteins [15]. OGT is conserved throughout eukaryotes, from *Caenorhabditis elegans* to mammals [8]. Unlike other cell hosts such as mammalian or insect cells, however, OGT has not been found in *E. coli* [16].

Although more than 4000 proteins (including many disease-related proteins) have been found to be O-GlcNAcylated, the function of O-GlcNAcylation in most proteins has not been clarified [17,18]. It is mainly attributed to the low abundance and small molecular weight of O-GlcNAcylation, and unstable O-glycosidic linkage. Preparing O-GlcNAc peptide/protein is essential to elucidate the role of specific GlcNAc-based pathways. Chemical and biochemical methods have been developed to produce O-GlcNAc peptide/protein [19].

At present, the synthesis methods of O-GlcNAc-modified peptide/protein mainly include OGT co-expression modified protein in *E. coli* and the chemical synthesis of modified peptide. The chemical synthesis of O-GlcNAc peptide/protein, selectively adding GlcNAc to specific sites of target peptides/proteins using solid phase or chemoenzymatic synthesis, is widely used to generate a specific antigen and desired antibody by immunizing animals [20], such as c-Myc (Thr58) [21], histone H2A (Ser40, Thr101) [22], Tau (Ser400) [23,24], histone H2B (Ser112) [25], SIRT1 (Ser549) [26], insulin receptor substrate 1 (Ser1011) [27], TAB1 (Ser395) [28], and TAB3 (Ser408) [29].

The co-expression of human OGT with a substrate protein in *E. coli* is a strategy to produce milligram-scale quantities of O-GlcNAc protein, including Nup62, CaMKIV, CARM1, CKII, p53, TAB1, H2B, Tau, and others. Shen et al. used O-GlcNAc proteins prepared in *E. coli* to measure the activity of human OGA in diverse protein substrates TAB1, CaMKIV, CARM1, Tau, and Nup62 [30]. Han et al. established an adjustable compatible dual plasmid system with varying O-GlcNAc levels by altering the inducer concentration, which could effectively yield O-GlcNAc CKII and p53 proteins in *E. coli* [31]. Gao et al. further optimized by co-expressing GlmM/GlmU in *E. coli*, which promoted the synthesis of UDP-GlcNAc to increase OGT activity, and effectively improved the O-GlcNAc level of target proteins TAB1, H2B, and p53 [32]. Yuzwa et al. prepared a Tau S400-O-GlcNAc site-specific antibody by immunizing animals with glycopeptides as the antigen and verified this specific antibody by co-expressing the Tau protein with OGT in *E. coli* [24]. Obtaining proteins with a high O-GlcNAc modification level is the key to the preparation of monoclonal antibodies and further functional research. However, the chemical synthesis of O-GlcNAc peptide/protein is difficult and costly, and biosynthesis needs to increase the proportion of O-GlcNAcylation [19].

In this study, an OGT binding peptide (OBP)-tagged strategy was developed to increase the O-GlcNAc level of a protein through the co-expression of OGT and tagged protein. This strategy successfully improved the O-GlcNAc level of Tau, H2B, and c-Myc proteins, which was far more efficient than control proteins without P1 and could conveniently and efficiently produce milligram-scale quantities of O-GlcNAc protein with a high O-GlcNAc level. Compared with Tau, O-GlcNAc of P1Tau had obvious interference with P1Tau aggregation in vitro. This strategy provides convenience for further preparation of a large number of O-GlcNAc-modified proteins and functional research.

## 2. Results

### 2.1. OBPs Increased O-GlcNAc Level of Tau

It is speculated that OGT has a higher catalytic efficiency for proteins with higher affinity. To improve the affinity between OGT and its substrate, a fusion strategy of OBP and target protein was designed to improve the O-GlcNAc level of the target protein. According to the research of Pathak et al., six OBPs were identified by using a substrate library of synthetic peptides [33]. From the above six peptides, the peptide KKVPVSRA with the highest binding activity to OGT, named P1, and the other two peptides with lower binding activity, KKVPVTRA and KKVGVSRA, named P2 and P3, were selected. Tau was selected as the target protein, and its O-GlcNAc site and function have been clearly studied in previous reports [34,35,36,37]. Then, P1, P2, and P3 were used as tags to fuse with the substrate Tau, named tagged Tau (P1Tau, P2Tau, P3Tau) (Figure 1A). OGT and the target protein (Tau, P1Tau, P2Tau, P3Tau) were cloned into vector pET-4CDS, which could express Tau or tagged Tau and OGT simultaneously in *E. coli* (Figure 1B). As shown in Figure 1C, the O-GlcNAc level of P1Tau was 4~6-fold higher than Tau, and the O-GlcNAc level of P2Tau and P3Tau was 2~3-fold higher than Tau, indicating that the fusion OBP (P1, P2, or P3) could increase the O-GlcNAc level of Tau. The O-GlcNAcylation of P1Tau was the highest among the tagged Tau, which was consistent with the highest affinity activity of P1 with OGT among the three OBPs (Figure 1D) [33].

To investigate the positional effect of tag, P1 was constructed at the N-Terminal (P1Tau) or C-terminal (TauP1) of Tau, respectively. The protein expression level of P1Tau (or TauP1) and O-GlcNAc level of P1Tau (or TauP1) were detected in the lysis of *E. coli* using anti-His antibody and RL2 antibody, respectively. The normalized relative ratio of Tau O-GlcNAcylation showed that the O-GlcNAc levels of both P1Tau and TauP1 were significantly enhanced 4~6-fold compared with Tau. Its level was slightly higher in TauP1 than in P1Tau, without a significant difference. (Figure 1E,F).

### 2.2. O-GlcNAc Sites Identified in Tau and Tagged Tau

The O-GlcNAc Tau protein was purified under different induction temperatures. It was found that P1Tau and TauP1 were all expressed well under 16 °C, 25 °C, or 37 °C (Appendix AA, Appendix A). Then, the recombinant O-GlcNAc-modified Tau, P1Tau, and TauP1 were purified using Ni-NTA columns (Appendix AB).

Tau, P1Tau, and TauP1 were purified using Ni-NTA, and the O-GlcNAc sites were detected using HCD/ETD. The array of low-mass oxocarbenium ions (*m*/*z* 126.06, 138.06, 144.07, 168.07, 186.08, and 204.09) generated upon HCD were characteristic of N-acetylglucosamine (GlcNAc). GlcNAc oxonium ion provided evidence of an O-GlcNAc-modified peptide. Based on the mass difference of 203.08 Da in the HCD/ETD spectra, we identified nineteen O-GlcNAcylated sites on Tau, seven sites on P1Tau, and eight sites on TauP1. The locations of these O-GlcNAc sites in the Tau, P1Tau, and TauP1 proteins are shown in Figure 2A. Interestingly, P1Tau displayed a similar modification pattern to TauP1 in the Proline-rich and C-terminal domain. These results indicated that P1 had little effect on O-GlcNAc-modified sites at either the N-terminal or C-terminal.

As shown in Table 1, a total of twenty-two O-GlcNAc modification sites were identified using mass spectrometry (MS), including nine previously reported ones and thirteen new ones. The new O-GlcNAc site S184 (Figure 2B) was identified in both Tau and tagged Tau. S400/T403 indicates O-GlcNAc at one of the sites. T386 (Figure 2C) and S400/T403 (Appendix AA) were identified in Tau and tagged Tau, and T76 (Appendix AB) and S305 (Figure 2D) were identified in tagged TauP1. Another eight new sites, T181 (Figure 2E), S195 (Appendix AC), S198, S241/T245 (Appendix AD), S316 (Appendix AE), T414, S416 (Figure 2F), and T427 (Appendix AF), were only identified in Tau. Of the nine previously reported sites identified, S185, S191, S396, and S400 were identified in both Tau and tagged Tau (Appendix AA, Appendix AB, Appendix AC, Appendix AD), and T205, S208, S412, S413, and S422 were only identified in Tau (Appendix AE,F, partial spectrum). These results indicated that nineteen out of the twenty-two O-GlcNAc sites were identified in Tau. However, far fewer O-GlcNAc sites were identified in tagged Tau than Tau, most of which were common sites in all samples. Combined with the results of Western blotting (Figure 1E), the P1-tagged strategy could promote the interaction between OGT and specific target proteins, improve the O-GlcNAc level of protein, and reduce the diversity of modified sites, which provides materials for subsequent antibody preparation and functional research.

### 2.3. P1Tau Aggregated More Slowly than Tau

As shown above, the O-GlcNAc modification level of the Tau protein was significantly enhanced in the presence of P1. The positive control Tau protein (no OGT) without OGT co-expression, Tau, and P1Tau protein with OGT co-expression were purified using SP cation exchange chromatography for aggregation experiment (Figure 3A). Another anti-O-GlcNAc antibody, CTD110.6, was used to detect the O-GlcNAcylation of Tau and P1Tau. The results showed that the O-GlcNAc level of P1Tau was higher than that of Tau (Figure 3B). To test the effect of increased O-GlcNAcylation on the aggregation rate, we conducted a kinetic analysis of Tau (no OGT), Tau, and P1Tau aggregation in vitro with Congo red and ThT binding assays.

The absorbance at 550 nm represents the amount of Tau fibrils to monitor the kinetic effect on the formation of Tau protein amyloid fibers induced by Congo red. Tau (no OGT) was used as a positive control and HEPES buffer as a negative control. The empirical Hill equation was fitted to the kinetic data in Figure 3C, and the solid lines represent the best fit. The normalized platform value of Tau (no OGT) fiber growth curve showed that Tau aggregated more slowly than Tau (no OGT) and its equilibrium (plateau) value decreased by ~82%, while the equilibrium value of P1Tau decreased by ~50%. To assess the potential effect of P1 on the Tau aggregation, Tau (no OGT) was incubated in HEPES buffer with or without peptide P1. As shown in Figure 3D, P1 had no obvious effect on Tau fibers’ aggregation and fluorescence reaction.

The heparin induced Tau protein aggregation, which was monitored through fluorescence spectroscopy using Thioflavin T (ThT) [38]. Tau protein aggregation occurs in the sigmoid shape curve with a readily apparent lag phase. Three kinetic parameters were obtained from the fitted sigmoid equation, as summarized in Table 2. As shown in Figure 3E, the aggregation of both Tau and P1Tau have an obvious lag phase compared with Tau (no OGT), and P1Tau exhibited a significantly longer lag time of (233.1 ± 4.8) min than that Tau of (138.9 ± 2.2) min.

Overall, these results confirmed that increasing the O-GlcNAc level of P1Tau significantly slowed the formation of amyloid fibrils during Tau protein aggregation.

### 2.4. P1 Tag also Improved the O-GlcNAc Level of c-Myc and H2B

To further verify the feasibility of P1-tagged strategy in improving the O-GlcNAc level of substrate protein, the other two proteins Myc and H2B were each fused with P1. P1 was constructed into the N-terminal or C-terminal of the target protein, respectively. The protein expression of tagged Myc (P1Myc or MycP1) and the O-GlcNAcylation of tagged Myc in the lysis of *E. coli* were detected using anti-His antibody and RL2 antibody, respectively (Figure 4A). Compared with the control group Myc, the O-GlcNAc level of tagged Myc showed a 17~21-fold increase (Figure 4B). Similarly, the protein expression of tagged H2B (P1H2B or H2BP1) and the O-GlcNAcylation of tagged H2B were detected using anti-His antibody and anti-H2B S112 O-GlcNAc antibody, respectively (Figure 4C). Compared with the control group H2B, the O-GlcNAc level of tagged H2B increased ~3-fold (Figure 4D). P1 showed similar results at the N-terminal or C-terminal of the protein without positional effect. Taken together, these results demonstrated that this strategy successfully expressed O-GlcNAc proteins with a far higher O-GlcNAc level than the control group Myc (H2B) and OGT co-expression.

## 3. Discussion

In this paper, the OBP-tagged strategy was used to successfully improve the O-GlcNAc level of Tau, H2B, and c-Myc proteins. It has been reported that the co-expression of OGT with its target substrates in *E. coli* could produce O-GlcNAc recombinant proteins using the dual-plasmid system [24,31,32]. However, the transfection efficiency of two plasmids in one system is different, and there is a possibility of plasmid loss. In this study, a new OBP-tagged strategy was developed to address the above problems successfully. The advantage of this strategy is that the target gene and OGT gene were cloned into a plasmid (pET-4CDS), which ensures the consistency of transfection efficiency and avoids the inconvenience and the loss of plasmids in a dual-plasmid system.

Although P1 tag is the substrate of OGT, the O-GlcNAc modification of P1 depends on the fusion proteins. The Ser on P1 in P1p53 was O-GlcNAcylation, but not in P1Tau. These data indicated that the function of the P1 tag is to promote the binding of OGT and the substrate protein, rather than compete for O-GlcNAc sites of the substrate. The OBP-tagged strategy with a high affinity of OGT is a successful approach. We will attempt to find more peptides with higher affinity for OGT to further optimize this OBP-tagged strategy. Ramirez et al. reported that OGT fused with tag-bound nanobody could enhance the O-GlcNAc level of the target protein in HEK293T cells [39], but this was not attempted in *E. coli*. Therefore, the effectiveness of this tag-bound nanobody in producing O-GlcNAc-modified protein is worthy of further study and verification in *E. coli*. Previous studies have implied that the asparagines in the N-terminal superhelical tetratricopeptide repeat (TPR) lumen proximal to the catalytic domain of OGT play a critical role in the recognition of most substrates [40,41,42]. However, this recognition has a preference for the amino acid composition of the substrate [43]. The amino acid preferences play an important role in improving binding to TPR, which has a reference significance for designing the TPR recognition tag. The tag P1 selected in this study is an OGT catalytic region binding peptide, while other peptides with a high affinity to OGT-TPR used as tags may be more worthy in an OBP-tagged strategy. Additionally, the O-GlcNAc-modified sites resulting from the OGT-TPR binding tags will be further compared with the catalytic region binding P1.

The results of Tau and tagged Tau O-GlcNAc sites showed that this strategy not only improved the O-GlcNAc modification level, but also reduced the diversity and microheterogeneity of O-GlcNAc sites, which may be more similar to the intracellular modification status in the microenvironment. S400 is an important O-GlcNAcylated site in the Tau protein that plays a major role in regulating Tau aggregation because the S400A mutation in recombinant O-GlcNAcylated Tau eliminates the inhibition of wild-type O-GlcNAcylation on Tau aggregation in vitro [34,35]. The S400 O-GlcNAc site was identified in all three samples of Tau and tagged Tau. S184 is a new site with high reliability, which deserves further study. Except for the S400 site-specific antibody [24], no antibodies for other specific modified sites of Tau were prepared. P1Tau and TauP1 may be used as antigens for preparing the S184-O-GlcNAc site-specific antibody and functional research. These results lay a foundation for the preparation of other site-specific antibodies against H2B and c-Myc proteins and the further study of their functions. 

Many O-GlcNAc-modified proteins have been identified with multiple O-GlcNAc sites, some of which have important functions [2,37]. Mutations in functional O-GlcNAc sites are always accompanied by a significant reduction in the O-GlcNAc level of these proteins [44,45,46]. The large-scale preparation of proteins with functional O-GlcNAc sites in vitro is of great significance for functional research. In this study, the results of the Tau protein indicated that large amounts of proteins with functional O-GlcNAc sites could be expressed and purified in vitro. However, the effectiveness of this strategy still needs to be verified using more proteins.

In conclusion, we report an in vitro method for enhancing protein O-GlcNAcylation using the OBP-tagged strategy, which provides highly modified and homogeneous samples for studying the biological functions of protein O-GlcNAcylation.

## 4. Materials and Methods

### 4.1. Preparing Gene Constructs in pET-4CDS

The OGT target proteins used in this study were human Tau441, H2B, and c-Myc. Taking the P1Tau fusion gene as an example, we constructed a fusion gene of P1 and Tau. The flexible and non-helical linker (GSSSS)_2_ was inserted between P1 and Tau. Using the Tau gene as the template, a pair of primers, P1Tau-Fa/Tau-R (P1Tau-Fa: CTC GTG CGG GCG GCG GCG GCA GCG GCG GCG GCG GCA GCG CTG AGC CCC GCC AGG AG and Tau-R: GGT GGT GGT GGT GCT CGA GCT GCA GCA AAC CCT GCT TGG CCA G), were designed for amplification. Then, using the above PCR product as a template, P1Tau-Fb/Tau-R (P1Tau-Fb: AAG GAG ATA TAC CAT GGT CCA TAT GAA AAA GGT GCC GGT CTC TCG TGC GGG CGG CGG CG and Tau-R: GGT GGT GGT GGT GCT CGA GCT GCA GCA AAC CCT GCT TGG CCA G) were designed for the second amplification. This second PCR product was inserted into the pET-4CDS vector between Nde1 and Pst1. A pair of PCR primers, OGT-F/R (OGT-F: GCG GAT GGG ATC CGA ATT CGT CGA CAT GAA AAT CGA AGA AGG TAA AC and OGT-R: TGC CCA CGG AAG ACG CCA TGT CGA CGA TAT CGC GGC CGC CCA TGG), were designed to amplify the OGT gene. The PCR product was inserted into pET-4CDS at the Sal1 site (restriction enzymes were purchased from Takara). The pET-4CDS was given by Dr. Yanting Su. Amplification of the genes was carried out using high-fidelity polymerase enzyme 2×Phanta Flash Master Mix (Vazyme). 2×MultiF Seamless Assembly Mix (Abclonal) was used to link genes with vectors. The final constructs were verified through DNA sequence analysis.

### 4.2. Production of O-GlcNAcylated Protein

The above plasmids were transformed into *E. coli* BL21 (DE3) prepared in our laboratory. The bacterial strains were grown in LB containing kanamycin (50 μg/mL) at 37 °C until OD 600 reached 0.6~0.8. Then, the culture was induced with 1 mM IPTG and further incubated at 25 °C for 12 h. The bacterial solution after induction was centrifuged at 3500 rpm for 25 min, and the supernatant was discarded.

#### 4.2.1. Ni-NTA Column

The remaining sediment was resuspended in binding buffer (50 mM sodium phosphate, 500 mM NaCl, 20 mM imidazole, pH 7.4) and broken ultrasonically. After centrifugation (9000 rpm, 25 min), the supernatant was purified as recombinant protein using a Ni-NTA affinity column (TransGen Biotech). The target proteins adsorbed on the Ni-NTA affinity column were eluted with an elution buffer (50 mM sodium phosphate, 500 mM NaCl, 200 mM imidazole, pH 7.4), and the protein sample was added to 5× loading buffer, boiled at 100 °C for 10 min, and separated in SDS-PAGE for LC-MS/MS analysis.

#### 4.2.2. SP Sepharose Column

The above bacteria were harvested and resuspended in SP buffer (20 mM Na_2_HPO_4_·12H_2_O, 20 mM NaH_2_PO_4_·2H_2_O, 1 mM EDTA, 0.5% β-Mercaptoethanol, 1 mM PMSF, pH 6.8) and were then broken by sonication on ice. The supernatants were filtered, loaded onto a cation exchanger HiTrap SP FF (GE Healthcare), and washed with SP buffer. A linear gradient of salt (0–400 mM NaCl) in the same buffer as used to elute the Tau protein. BCA Protein Assay (Yeasen) and Coomassie brilliant blue staining of 10% SDS-PAGE were used to estimate protein concentration and purity, respectively. The purified proteins were dialyzed three times with 10 mM HEPES buffer (10 mM HEPES, 100 mM NaCI, 0.5% β-ME, pH7.4), concentrated with a 10K Amicon centrifugal filter (Millipore) for kinetic analysis, and transferred to −80 °C for long-term storage.

### 4.3. Congo Red Binding Assay

Congo red, widely employed as an inducer for Tau aggregation in vitro, can react specifically with β-sheet-rich amyloid fibrils, and the bound form displays a characteristic red shift in its maximum absorbance from 490 to 550 nm [47,48]. A fresh 5 mM CR (Sigma-Aldrich, USA) stock solution was prepared in HEPES buffer containing 100 mM NaCl (pH 7.4) and filtered (0.22 µm pore size) before use to remove insoluble particles. The polymerization induced by Congo red for full-length human Tau (no OGT) and its O-GlcNAcylated Tau in 96-well plates (Corning) was set up with a mixture of 10 μM proteins and 10 μM P1 peptide (synthesized by GenScript) incubated with 50 μM Congo red in 10 mM HEPES buffer containing 1 mM DTT (pH 7.4). To block the formation of an intramolecular disulfide bond in the Tau protein, 1 mM DTT was added into the HEPES buffer. The reaction components were mixed quickly and immediately read at 37 °C for 6 h in a Cytation 3 Microporous Plate Reader (BioTek, USA) using absorbance at 550 nm. All kinetic experiments were repeated three times. Kinetic parameters were determined by fitting the absorbance at 550 nm versus time to the empirical Hill equation [47]:(1)$$A=A\left(\infty \right)\frac{{\left(t/t_{50}\right)}^{n}}{1+{\left(t/t_{50}\right)}^{n}}$$
where *A* is the absorbance at 550 nm, *A*(∞) is the absorbance at 550 nm in the long time limit, *t*_50_ is the elapsed time at which *A* is equal to one-half of *A*(∞), and *n* is a cooperativity parameter.

### 4.4. ThT Binding Assays

The aggregation of Tau protein normally occurs through the nucleation-dependent fibril polymerization in three phases: activation and nucleation (Lag phase), elongation (growth phase), and steady phase (paired helical filaments, PHFs) [49]. A small fluorescent molecule, Thioflavin T (ThT), can be bound preferentially to amyloid fibrils when free in solution, which has been frequently used for monitoring the kinetics of amyloid fibril formation [50]. A stock solution of 2.5 mM Thioflavin T (ThT) (Sigma-Aldrich, St. Louis, MO, USA) was freshly prepared in 10 mM HEPES buffer. Recombinant Tau proteins (10 μM) were incubated in 10 mM HEPES buffer containing 1 mM DTT and 25 μM ThT at 200 rpm and 37 °C for 10 h in the presence of 2.5 μM heparin. The resulting 200 µL mixtures were added to a 96-well fluorescence microplate (Corning) in a Cytation 3 with an excitation wavelength of 440 nm and emission wavelength of 480 nm. Measurements were made at 37 °C with agitation. All kinetic experiments were repeated three times. Kinetic parameters were determined by fitting ThT fluorescence intensity versus time to a sigmoidal equation [51]:(2)$$F=F_{0}+\left(A+ct\right)/\left\{1+e^{\left[k\left(tm-t\right)\right]}\right\}$$
where *F* is the fluorescence intensity, *k* the rate constant for the growth of fibrils, and *tm* is the time to 50% of maximal fluorescence. *F*_0_ describes the initial baseline during the lag time. *A + ct* describes the final baseline after the growth phase has ended. The lag time is determined to be *tm −* 2/*k*.

### 4.5. SDS-PAGE and Western Blot

The target proteins were fractionated with SDS-PAGE (Tau and Myc, 10%; H2B, 15%; OGT, 8%) and visualized by staining with Coomassie blue. For Western blot analysis, the proteins were transferred to PVDF membrane (Millipore, Burlington, MA, USA), and blocked with 5% non-fat milk (or bovine serum albumin) for 1 h at room temperature or overnight at 4 °C, after separating on SDS-PAGE. Target proteins Tau, H2B, and Myc were identified by using anti-His antibody conjugated to HRP (1:10000) (Abclonal, Woburn, MA, USA). Likewise, the OGT was detected by using mouse anti-Flag (1:1000) (Abclonal) as the primary antibody and goat anti-mouse antibody conjugated to HRP (Abclonal) as the secondary antibody. The O-GlcNAcylation of the target was identified by using anti-O-GlcNAc antibody RL2 (Abcam, Cambridge, UK), CTD110.6 (Cell Signaling Technology, Danvers, MA, USA) and anti-H2B S112GlcNAc (Abcam). Super ECL Detection Reagent (Yeasen, Shanghai, China) was used for developing and capturing proteins. Blots were visualized using an automatic chemiluminescence image analysis system (Tanon 5200). The expression of proteins was analyzed using ImageJ and Graphpad prism 7.

### 4.6. Mass Spectrometry Analysis

SDS-PAGE-separated O-GlcNAcylated proteins Tau, P1Tau, and TauP1 were reduced using 10 mM dithiothreitol and alkylated in 55 mM iodoacetamide (Sigma-Aldrich, St. Louis, MO, USA) prior to in-gel proteolytic digestion with trypsin (1:50, enzyme: protein ratio) for 12 h. The digested peptides were extracted from the gel pieces by sequentially adding 0.1% formic acid in 50% acetonitrile, 0.1% formic acid in 80% acetonitrile, and 100% acetonitrile. The extraction products were combined and lyophilized for the following step. The MS approach employed in this study for sequencing O-GlcNAcylated peptides and assigning the modification sites is a combination of HCD and ETD fragmentation. When analyzing O-GlcNAcylated peptides, the unique advantage of HCD fragmentation is the generation of distinct HexNAc oxonium ions (*m*/*z* 204.09) and a series of HexNAc fragments (*m*/*z* 186.08, *m*/*z*168.07, *m*/*z*144.07, *m*/*z* 138.06, and *m*/*z*126.06), which are significant to the diagnosis of O-GlcNAcylated peptides. The samples were analyzed using Fourier transform ion cyclotron resonance (FT-ICR/Orbitrap, Thermo Fisher Scientific, San Jose, CA, USA). The FT-ICR/Orbitrap mass spectrometer was equipped with an online nano-electrospray ionization (ESI) source. The database search results were imported into Peaks Studio 10.6. Search parameters included a precursor tolerance of 10 ppm and a fragment tolerance of 0.02 Da. Enzyme specificity was set to trypsin, and up to two missed cleavage sites were allowed. Carbamidomethylation was set as a fixed modification. For the main GlcNAc search, HexNAc (S/T) was set as a variable modification. GlcNAc was considered positively localized to serine or threonine residues with FDR less than 1%. Putative O-GlcNAcylated peptides were manually confirmed by checking for the presence of the expected HexNAc neutral loss from the precursor ion and/or fragmented HexNAc oxonium ions. 

## Figures and Tables

**Figure 1 molecules-28-02129-f001:**
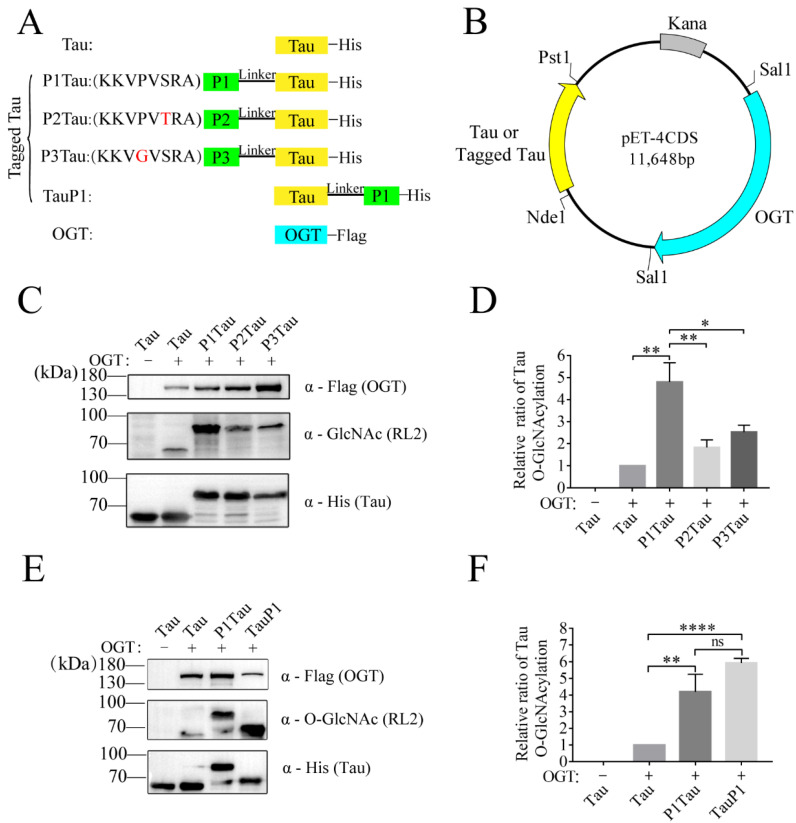
Tagged Tau increases O-GlcNAcylation of Tau protein in *E. coli*. (**A**) Linear representation of target gene and tag fusion genes. (**B**) Overview of the OGT binding peptides (OBP)-tagged strategy. (**C**,**E**) The expression levels of OGT, Tau (or tagged Tau,) and O-GlcNAc Tau (or tagged Tau) were analyzed using Western blot with anti-Flag, anti-His, and anti-O-GlcNAc antibody (RL2). (**D**,**F**) The relative ratio of Tau O-GlcNAcylation was normalized and is presented as mean ± S.D. (*n* = 3 biologic replicates). Statistical analyses were performed using Student’s *t* test, ****, *p* < 0.0001. **, *p* < 0.01. *, *p* < 0.05. ns, *p* > 0.05.

**Figure 2 molecules-28-02129-f002:**
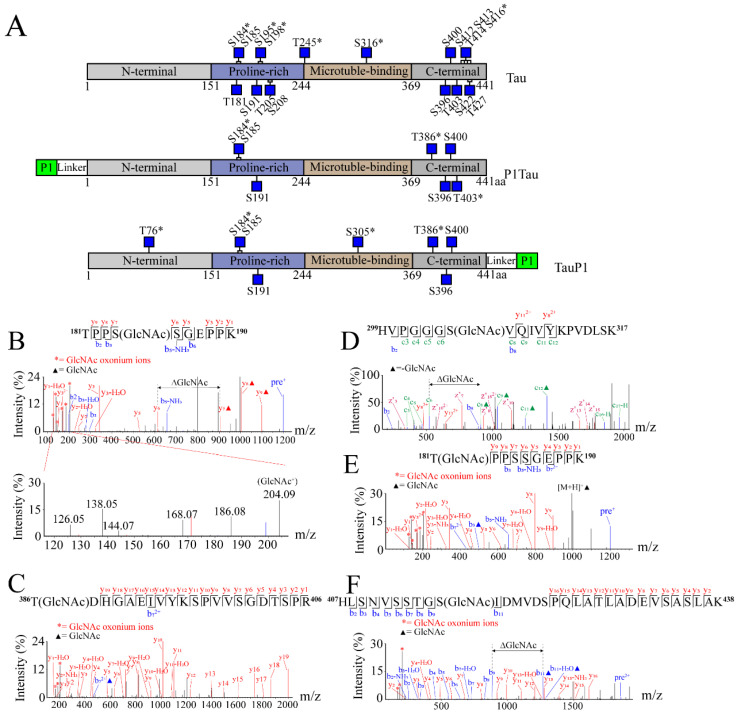
O-GlcNAc sites of Tau, P1Tau, and TauP1 were identified using MS. (**A**) Schematic structures of Tau, P1Tau, and TauP1 with O-GlcNAc-modified residues indicated as blue squares, and novel sites indicated with asterisks (*). HCD/ETD spectra reveal some new O-GlcNAc sites on the following peptides: (**B**) ^181^TPP**S**SGEPPK^190^ (P1Tau), GlcNAc oxonium ions (126.06, 138.06, 144.07, 168.07, 186.08 *m*/*z,* and 204.09 *m*/*z*) are shown in the lower panel of spectrogram; (**C**) ^386^**T**DHGAEIVYKSPVVSGDTSPR^406^ (P1Tau); (**D**) ^299^HVPGGG**S**VQIVYKPVDLSK^317^ (TauP1); (**E**) ^181^**T**PPSSGEPPK^190^ (Tau); (**F**) ^407^HLSNVSSTG**S**IDMVDSPQLATLADEVSASLAK^438^ (Tau). All of the modification sites are shown in underlined letters. Double-headed arrows and ΔGlcNAc indicate the mass shift of the O-GlcNAc moiety between key fragment ions. Oxonium ions are labeled with red asterisks. Solid triangles represent the presence of GlcNAc modification on the peptide ions and fragment ions.

**Figure 3 molecules-28-02129-f003:**
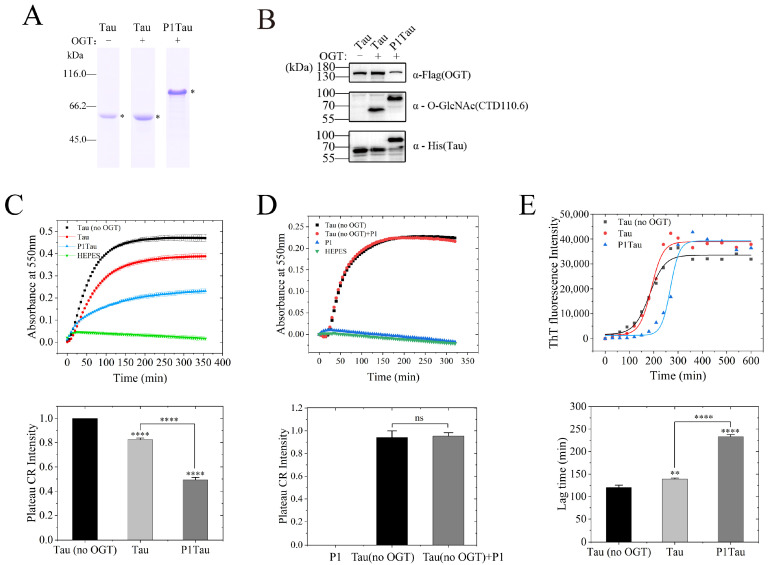
Increased O-GlcNAcylation dramatically decreased the fibrillization of full-length human Tau in vitro. (**A**) Coomassie blue (CB)-stained SDS-PAGE gel of Tau (no OGT), O-GlcNAcylated Tau, and P1Tau proteins were purified using a SP Sepharose column and analyzed using Coomassie brilliant blue staining. (**B**) The expression levels of OGT, Tau (or P1Tau), and O-GlcNAc Tau (or P1Tau) were analyzed using a Western blot with anti-Flag, anti-His, and anti-O-GlcNAc antibody (CTD110.6). (**C**) Samples (10 μM) of Tau (no OGT, black solid square), Tau (red solid circle), and P1Tau (blue solid triangle) aggregated using 50 μM Congo red (CR) and detected at 550 nm for 6 h. (**D**) Samples (10 μM) of Tau (no OGT, black solid square), Tau (no OGT) +P1 (10 μM) (red solid circle), and P1 (blue solid triangle) aggregated using 50 μM Congo red (CR) and detected at 550 nm for 6 h. The normalized plateau value of Tau fiber growth curve at the bottom determined a ratio of insoluble Tau aggregate in total Tau and is expressed as the mean ± S.D. (*n* = 3 biologic replicates). (**E**) Aggregation of 10 μM proteins in the presence of 2.5 μM heparin was detected using ThT fluorescence at 480 nm for 10 h. The lag times of fibril formation of Tau (no OGT), O-GlcNAcylated Tau, and P1Tau were determined using a sigmoidal equation using the data. The ThT fluorescence intensity and lag time are expressed as the mean ± S.D. (*n* = 3 biologic replicates). Statistical analyses were performed using Student’s *t* test, ****, *p*< 0.0001. **, *p* < 0.001. ns, *p* > 0.05.

**Figure 4 molecules-28-02129-f004:**
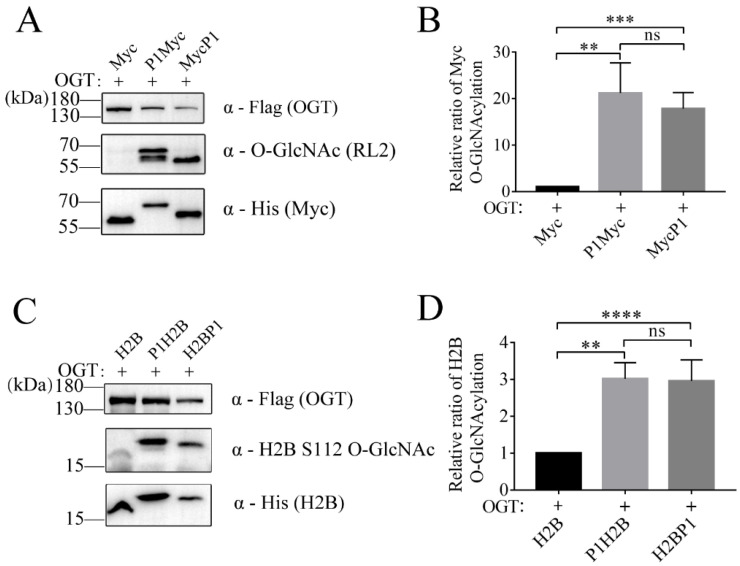
Targeting other proteins for O-GlcNAcylation with OBP-tagged strategy. (**A**) The expression and O-GlcNAc levels of Myc and (**C**) H2B were analyzed with a Western blot with anti-His, anti-Flag, and anti-O-GlcNAc antibody. The relative ratio of (**B**) Myc and (**D**) H2B O-GlcNAcylation was normalized and is presented as mean ± S.D. (*n* = 3 biologic replicates). Statistical analyses were performed using Student’s *t* test, ****, *p* < 0.0001. ***, *p*< 0.001. **, *p* < 0.01. ns, *p* > 0.05.

**Table 1 molecules-28-02129-t001:** O-GlcNAc sites of Tau and tagged Tau.

O-GlcNAc Sites	P1Tau	TauP1	Tau
T76 *		+	
T181 *			+++
**S184 ***	+++	+++	+++
**S185**	+++	+++	+++
**S191**	+++	+++	+
S195 *			++
S198 *			++
T205			++
S208			+++
S241/T245 *			+
S305 *		+++	
S316 *			+
**T386 ***	+	+	
**S396**	+++	+++	+++
**S400**	+++	+++	+++
**T403 ***			+
**S400/T403 ***	+		
S412			+++
S413			+++
T414 *			+
S416 *			+++
S422			++
T427 *			++

*: new O-GlcNAc sites. /: O-GlcNAc at one of the sites (S400/T403, S241/T245). +++: oxocarbenium ions (*m*/*z* 204.09, 186.08, 168.07, 144.07, 138.06, and 126.06) generated upon HCD characteristic of GlcNAc, and neutral loss or c/z ions produced by ETD near the amino acid (S/T) of O-GlcNAcylation; ++: oxocarbenium ions and neutral loss of ions; +: oxocarbenium ions or neutral loss of ions (±0.02 Da).

**Table 2 molecules-28-02129-t002:** Kinetic parameters of Tau and Tau-related protein aggregation in ThT binding assays.

Protein	*A + ct*	Lag Time (min)	*k* (h^−1^)
Tau (no OGT)	31935.7 ± 1678.8	120.3 ± 5.3	1.99 ± 0.35
Tau	37526.3 ± 1634.5	138.9 ± 2.2	2.36 ± 0.36
P1Tau	37937.9 ± 1709.7	233.1 ± 4.8	3.30 ± 0.61

## Data Availability

Not applicable.

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
