# Peer review of "OGT Binding Peptide-Tagged Strategy Increases Protein O-GlcNAcylation Level in E. coli"

_molecules, 2023, doi:10.3390/molecules28052129_

Round 1
Reviewer 1 Report
The O-GlcNAc post-translational modification (PTM) involves the reversible attachment of single monosaccharide, N-Acetylglucosamine (GlcNAc), units onto serine and threonine residues of nucleocytoplasmic proteins. O-GlcNAc is added by the action of OGT, a glycosyltransferase, and removed by O-GlcNAcase, a glycoside hydrolase. O-GlcNAc is added to a diverse array of eukaryotic proteins and has been revealed to have numerous functional roles on such proteins. One major limitation, however, in this field is the lack of understanding of the function of individual O-GlcNAc modification sites on different proteins. This limitation has arisen due to the lack of site-specific O-GlcNAc modification antibodies (most antibodies recognize O-GlcNAc only and not the site-specific context) and the inability to generate O-GlcNAc modified proteins to study in vitro and in vivo. This manuscript, by Li et al, aims to address the latter issue by improving on an E.coli dual expression system to boost the amount of O-GlcNAc on modified proteins so this protein can then be used in functional assays or in theory be used to raise O-GlcNAc site specific antibodies. These authors, make use OGT binding peptides previously described and fuse these to either the C- or N-Terminus of Tau, c-Myc or H2B and then co-expressed the tagged proteins with OGT using a single plasmid approach. The microtubule-associated protein tau (hereafter tau) makes up one half of the pathology of Alzheimer’s Disease when it forms aggregates. Tau aggregation has previously been shown to be negatively impacted by O-GlcNAc and thus is a potential therapeutic target. Li et al. make their major focus on the production of tau with enhanced O-GlcNAc modification. Using Western blotting they show P1tau and TauP1 have 4-6 higher O-GlcNAc levels, identify O-GlcNAc sites by mass spectrometry (MS) and show that P1 tau has lower aggregation. Finally, these authors show that tagged c-Myc and H2B also have enhanced O-GlcNAc modification.
While I find this work interesting and generally well done, I would like to raise a number of issues that I think should be addressed:
1 1)The enhanced site-specific O-GlcNAc on tagged tau compared to tau is very interesting but the significance of this is not assessed. Can it be ruled out that the tagged tag biases where OGT binds on tau and thus limits the sites that it modifies? Related to this, I would like to see an assessment of tagged tau, other than Western blotting, that shows that tagged tau has enhanced O-GlcNAc. It is possible that there is simply enhanced RL2 antibody binding because of the site differences. The authors are already performing MS. The could simply compare the chromatograms (O-GlcNAc tau should elute earlier) to assess the degree of O-GlcNAc modification on tau in all cases as an orthogonal readout.
2 2)The P1 tag has the potential to change the function of tau and its long range interactions. This work would be improved by assessing whether the tag impacts tau specifically in the aggregation reactions and or microtubule polymerization assays (ie. tau’s main function).
3 3) Related to the above to confirm the aggregations phenotypes of tagged tau, it would be preferred to have an independent (non fluorescence) method to demonstrate this aggregation such as electron microscopy or filter-trap assay. This would eliminate any potential impacts of the tagged tau on the fluorescence based assays.
4 4) The OGT binding peptides used in this work from Pathak et al., are OGT substrate peptides (ie. could be O-GlcNAc modified). What is the contribution to the O-GlcNAc levels of these proteins purely from the tag itself?
5 5) I find the diagrams in Figure 1 of P1/P2/P3 tagged tau confusing as the tag is shown both as sequence and as a green schematic. The authors would improve the clarity of this by simplifying to one or the other.
6 6)There is limited discussion of how the statistical analysis was done. Were these three independent batches of these co-expressed proteins? Technical replicates? This is important to assess the robustness of this effect.
7 7) The P2 and P3 tagged tau proteins are not really described at all in the results and yet they’re included in the figures. This needs to be addressed.
8 8)Methodologically, it’s not clear how the purification of various tau constructs was done. Was Ni-NTA and the SP Sepharose done, or one of the other? Clarity could be improved here because this could impact the aggregation of tau depending on how the purification was done.
9 9)Finally, there are a number of grammatical issues that need to be resolved.
Reviewer 2 Report
The manuscript provides a new method for the in vitro expression of highly O-GlcNAcylated proteins. Results are clear and unambiguous. However the following points should be carefully addressed, english needs to be significantly improved and typos corrected.
MAJOR POINTS
1- The authors claim that they provide a new way to get high yield of the O-GlcNAcylated form of a protein target. They should specify in conclusion (L. 260) that it is an in vitro approach.
Could they specify more precisely the quantity of the O-GlcNAcylated protein target that is purified ? (and from what volume of bacteria culture ?)
2- L.218 : ″much higher O-GlcNAc level than previously reported strategy″ : Please indicate the references
3- The authors argue that providing a homogeneously O-GlcNAcylated protein would help in studying the biological functions of this PTM. (L. 260-261). But a minor O-GlcNAc site may be missing when it may be functionally important. Could the authors discuss on this point and may be provide arguments from published works.
4- L. 254 « indicating that the common site identified may be more important ». This idea would need to be justified. (quite similar remark to point -2) Have the authors identified the O-GlcNAc sites on P1-Myc and if so, was Thr58 identified ?
5- It is unclear from which purification process the MS experiments were performed. This should be clearly mentionned in the results. Could they also explain why they need to purify the recombinant O-GlcNAc protein by cation exchange rather than with Ni-NTA affinity column ? Could they provide the chromatogram and stained SDS-PAGE as supplementary data ?
6- L. 240 ″Therefore, this tag-bound nanobody to produce O-GlcNAc modified proteins also deserved to further try and verify its effectiveness in E. coli. The tag P1 selected in this study is an OGT catalytic region binding peptide, while other peptides with high affinity to OGT-TPR used as tags may be more worthy of attempting in OBP-tagged strategy. And the O-GlcNAc-modified sites result of the OGT-TPR binding tag will be further compared with the catalytic region binding P1 tag.″
These sentences should be revised.
OGT has a broad specificity, in particular coming from the binding of protein targets in the TPR domain. It would be interesting to discuss the latest findings on the importance of some residues in the TPR domain of OGT (such as in Levine et al, 2018 ; Joiner et al, 2019 ; Ramirez et al, 2021 ; .Chong et al, BioXRIv 2022,...,) How can they author rely these findings to the potential development of new linked peptides to apply their strategy ? Would there not be competition between the peptide and the protein target ?
7- How to explain the difference in the apparent MW of O-GlcNAc P1-Tau and Tau-P1 (Fig 1E). Was the same linker used for P1-Tau and Tau-P1 ?
8- FigS2A and S2D: There are very few fragment ions on the spectra. How can they be sure that T403 and T245 are the right O-GlcNAc sites, respectively ?
9- Is there any size limitation of protein target for such a strategy (coexpression with OGT on the same vector)?
MATERIAL AND METHODS. Some information is missing
- Please specify pET-4CDS vector’s and E. coli BL21 (DE3) strain’s suppliers and the species of the cloned sequences
- L. 304 The purified proteins were exchanged into 10 mM HEPES buffer. Please specify how it was performed. No protease inhibitors were added in the extraction and purification buffer ?
- L.310 please indicate the concentration of CR stock solution
- L. 328 please indicate what PHF means
- L. 352 which system was used for ECL imaging ?
- protocol for trypsin digestion (or a reference) and on which samples it was done.
- manufacturer for FT-ICR/Orbitrap
- L . 344 ″The target proteins were fractionated with different concentrations of SDS-PAGE″. The sentence should be revised.
TYPOS
Fig 1A : microTUBLES binding
L.44 have been found to be O-GLCNACYLATION
L.61 to MEASURED human O-GlcNAcase glycosidase ACTIVITIES in diverse protein substrates...
L.87 six OGT binding PEPTIDE were identified by…
L.140 S185, S191, S396 and S400 were IDENTIFID in both
L.207 P1 was constructed into the N-terminal or C-terminal of target PTOTEIN, respectively
L. 254 identified common SITE
L. 253-254 : please harmonize the way ″Tau″ is written throughout the manuscript
L.258 O-GcNAc site
L. 349 by using HRP-anti-His antibody conjugated to HRP
Fig2B-F : please insert a space between y-axis legend and bracket: Intensity (%)
Table 2 : please align numbers in the last column
ENGLISH needs to be significantly improved (for ex., L16-17, L30, L45, L67, L123, L145,…L.293, … )
FIGURES
- Please indicate the position of the Flag on OGT (N-ter or C-ter ?). The Flag and His tags should be added on Fig 1
- Please check the MW standards for OGT WBlot in Fig1C-E and Fig 4A (see FigS1A)
- Please check for missing triangles on the peptide ions and fragment ions on spectra in Fig2SB, S2F,S2E
Round 2
Reviewer 1 Report
Overall, this manuscript is significantly improved and the authors have addressed a number of my concerns. I will, however, reiterate that doing a whole protein MS analysis or a western blot with a second O-GlcNAc antibody (such as CTD110.6) would clarify that total O-GlcNAc on tau is increased.
